# Non-convex Optimization for Learning a Fair Predictor under Equalized Loss Fairness Constraint

## Abstract

Supervised learning models have been increasingly used in various domains such as lending, college admission, natural language processing, face recognition, etc. These models may inherit pre-existing biases from training datasets and exhibit discrimination against protected social groups. Various fairness notions have been introduced to address fairness issues. In general, finding a fair predictor leads to a constrained optimization problem, and depending on the fairness notion, it may be non-convex. In this work, we focus on Equalized Loss (EL), a fairness notion that requires the prediction error/loss to be equalized across different demographic groups. Imposing this constraint to the learning process leads to a non-convex optimization problem even if the loss function is convex. We introduce algorithms that can leverage off-the-shelf convex programming tools and efficiently find the *global* optimum of this non-convex problem. In particular, we first propose the `ELminimizer` algorithm, which finds the optimal EL fair predictor by reducing the non-convex optimization problem to a sequence of convex constrained optimizations. We then propose a simple algorithm that is computationally more efficient compared to `ELminimizer` and finds a sub-optimal EL fair predictor using *unconstrained* convex programming tools. Experiments on real-world data show the effectiveness of our algorithms.

## 1 Introduction

As machine learning (ML) algorithms are increasingly being used in applications such as education, lending, recruitment, healthcare, criminal justice, etc., there is a growing concern that the algorithms may exhibit discrimination against protected population groups. For example, speech recognition products such as Google Home and Amazon Alexa were shown to have accent bias (Harwell, 2018). The COMPAS recidivism prediction tool, used by courts in the US in parole decisions, has been shown to have a substantially higher false positive rate for African Americans compared to the general population (Dressel & Farid, 2018). Amazon had been using automated software since 2014 to assess applicants' resumes, which were found to be biased against women (Dastin, 2018).

Various fairness notions have been proposed in the literature to measure and remedy the biases in ML systems; they can be roughly classified into two classes: 1) *individual fairness* focuses on the equity at individual level and it requires the similar individuals to be treated similarly (Dwork et al., 2012; Biega et al., 2018; Jung et al., 2019; Gupta & Kamble, 2019); 2) *group fairness* requires certain statistical measures to be (approximately) equalized across different groups distinguished by some sensitive attributes. Their suitability for use is often application dependent, and many of them are incompatible with each other (Zhang et al., 2019; Hardt et al., 2016; Conitzer et al., 2019; Zhang et al., 2020; Khalili et al., 2020). Extensive approaches have been developed to satisfying a given definition of fairness and they generally fall under three categories: *pre-processing*, by modifying the original dataset such as removing certain features and reweighing, e.g., (Kamiran & Calders, 2012; Celis et al., 2020); *in-processing*, by modifying the algorithms such as imposing fairness constraints or changing objective functions, e.g., (Zhang et al., 2018; Agarwal et al., 2018; 2019; Reimers et al., 2021; Calmon et al., 2017); *post-processing*, by adjusting the output of the algorithms based on sensitive attributes, e.g., (Hardt et al., 2016).

In this paper, we focus on group fairness and we aim to mitigate unfairness issues in supervised learning using in-processing approaches. The problem can be cast as a constrained optimization problem where a fair predictor can be found by minimizing the prediction error (i.e., loss) subject to certain group fairness constraint. In Section 2.1, we present a number of definitions of commonly used group fairness notions, namely, statistical parity (Dwork et al., 2012), equal opportunity (Hardt et al., 2016), equalized loss (Zhang et al., 2019), and bounded group loss (Agarwal et al., 2019). Here we are particularly interested in equalized loss which requires the expected loss to be equalized across different groups.

Constrained optimization problems for finding a fair predictor have been studied in the iterature. In general, imposing a fairness criterion to the optimization problem may lead to a non-convex optimization problem. Existing works have proposed various approaches to solving such a non-convex optimization in different settings. For example, Komiyama et al. (2018) studied the non-convex optimization for regression problems under the coefficient of determination constraint. Agarwal et al. (2019) proposed an approach to finding a fair regression model under bounded group loss and statistical parity fairness constraints. Agarwal et al. (2018) studied classification problems and aimed at finding fair classifiers under various fairness notions including statistical parity and equal opportunity. In particular, they considered zero-one loss as the objective function and trained a *randomized* fair classifier over a finite hypothesis space; this problem was reduced to a problem of finding the saddle point of a linear Lagrangian function in (Agarwal et al., 2018). Zhang et al. (2018) proposed an adversarial debasing technique to find a fair classifier under equalized odd, equal opportunity, and statistical parity. However, there is no guarantee that this technique finds the global optimal solution. The main difference between the present work and the existing in-processing approaches are as follows: 1) we consider a non-convex problem for finding a fair predictor satisfying *Equalized Loss* fairness notion, which has not been studied in the literature to the best of our knowledge. 2) We propose algorithms for finding the global optimal solution to this non-convex problem efficiently. 3) Our algorithms are easy to implement and are applicable to both regression and classification problems. 4) Unlike (Agarwal et al., 2018), our algorithms are not limited to finite hypothesis space.

Non-convex optimization problems have also been studied in other contexts such as learning over-parametrized models. For example, deep neural networks are typically trained by solving *unconstrained*, non-convex problems, and methods such as gradient descent may not be suitable as they are likely to find saddle points but not optimums. To address this issue, approaches have been proposed in recent works by incorporating the higher order derivatives (Celis et al., 2020; Anandkumar & Ge, 2016) or noisy gradients (Ge et al., 2015). However, these methods only find a local minimum (not a global minimum) and are not applicable to our problem with a non-convex constraint.

In this work, we develop novel algorithms that find the fair (sub-)optimal solutions under Equalized Loss fairness constraint efficiently. Note that while our approach and algorithms are presented in the context of fair machine learning, they are applicable to any problem that can be formulated as a constrained optimization problem in the form of $\min_{\boldsymbol{w}} L_0(\boldsymbol{w}) + \alpha L_1(\boldsymbol{w})$ s.t. $|L_0(\boldsymbol{w}) - L_1(\boldsymbol{w})| < \gamma$, where $\alpha$ is a constant

Our main contributions and findings are as follows.

1. We study the relationship between Equalized Loss (EL) and Bounded Group Loss (BGL) fairness notions. We show that given the existence of feasible solutions satisfying (approximate) BGL fairness, imposing (approximate) EL fairness constraint never increase losses of both groups simultaneously (Theorems 1 and 2 in Section 2.1). These results help policy makers to have a better understanding of these two fairness notions.

2. We develop an algorithm (ELminimizer) to solve a non-convex constrained optimization problem that finds the optimal (approximate) EL fair solution. We show that such non-convex optimization can be reduced to a sequence of *convex constrained* optimizations and the convergence property of the algorithm is analyzed (Theorems 3 and 4, Section 3).

3. We develop a simple algorithm for finding a *sub-optimal* (approximate) EL fair solution. We show that a sub-optimal solution is a linear combination of optimal solutions to two *unconstrained* optimizations and it can be found efficiently without solving constrained optimizations (Theorem 5, Section 4).

4. We conduct sample complexity analysis and provide the guarantee on generalization performance (Theorem 7, Section 5).

5. We validate the theoretical results by conducting experiments on real-world data (Section 6).

## 2 PROBLEM FORMULATION

Consider a supervised learning problem where the training dataset consists of triples $(\boldsymbol{X}, A, Y)$ from two social groups. Random variable $\boldsymbol{X} \in \mathcal{X} \subset \mathcal{R}^{d_x}$ is the feature vector (in form of a column vector), $A \in \{0, 1\}$ is the sensitive attribute (e.g., race, gender) indicating the group membership, and $Y \in \mathcal{Y} \subset \mathcal{R}$ is the label. The feature vector $\boldsymbol{X}$ may or may not include sensitive attribute $A$. Label $Y$ can be either discrete or continuous depending on the given problem: if $Y$ is discrete (resp. continuous), then the problem is a classification (resp. regression) problem. Let $\mathcal{F}$ be a set of predictors $f_{\boldsymbol{w}} : \mathcal{X} \to \mathcal{R}$ parameterized by weight vector $\boldsymbol{w} \in \mathcal{R}^{d_w}$.[1] Consider loss function $l : \mathcal{Y} \times \mathcal{X} \to \mathcal{R}$ where $l(Y, f_{\boldsymbol{w}}(\boldsymbol{X}))$ measures the error of $f_{\boldsymbol{w}}$ in predicting label $Y$. Denote the expected loss with respect to the joint probability distribution of $(\boldsymbol{X}, Y)$ by $L(\boldsymbol{w}) := \mathbb{E}\{l(Y, f_{\boldsymbol{w}}(\boldsymbol{X}))\}$. Then, $L_a(\boldsymbol{w}) := \mathbb{E}\{l(Y, f_{\boldsymbol{w}}(\boldsymbol{X})) | A = a\}$ denotes the expected loss of the group with attribute $A = a$.

A predictor that minimizes the total expected loss, i.e., $\arg\min_{\boldsymbol{w}} L(\boldsymbol{w})$, can be biased against certain groups. To mitigate the risk of unfairness, various fairness notions have been proposed in the literature. Some of the most commonly used notions of group fairness are as follows: 1) *Statistical Parity* (SP) (Dwork et al., 2012) implies that the predictor and the sensitive attribute should be independent, i.e., $f_{\boldsymbol{w}}(\boldsymbol{X}) \perp A$; 2) *Equal Opportunity* (EqOpt) (Hardt et al., 2016) requires that conditional on $Y = 1$, prediction and sensitive attribute are independent, i.e., $f_{\boldsymbol{w}}(\boldsymbol{X}) \perp A | Y = 1$; 3) *Equalized Odds* (EO) (Hardt et al., 2016) requires the conditional independence between prediction and sensitive attribute given $Y$, i.e., $f_{\boldsymbol{w}}(\boldsymbol{X}) \perp A | Y$; 4) *Equalized Loss* (EL) (Zhang et al., 2019; Berk et al., 2021) requires that the losses experienced by different groups are equalized, i.e., $L_0(\boldsymbol{w}) = L_1(\boldsymbol{w})$; 5) *Bounded Group Loss* (BGL) (Agarwal et al., 2019) requires that the loss experienced by each group is bounded.

With fairness consideration, the goal is to find weight vector $\boldsymbol{w}$ that minimizes total expected loss in predicting $Y$ given $\boldsymbol{X}$, subject to certain fairness condition, i.e., $\min_{\boldsymbol{w}} L(\boldsymbol{w})$ s.t. fairness constraint. This is a typical formulation in fair machine learning literature, and above method of finding a fair predictor belongs to *in-processing* approaches. Because such constrained optimization can be non-convex, finding the optimal solution efficiently can be challenging. In this work, we develop novel algorithms that solves such an optimization problem udder EL fairness constraint.

### 2.1 EQUALIZED LOSS (EL) AND BOUNDED GROUP LOSS (BGL)

As mentioned in Section 2, various fairness notions have been introduced in the literature. Among them, Statistical Parity (SP), Equal Opportunity (EqOpt), Equalized Odds (EO), and Bounded Group Loss (BGL) have been studied extensively in the literature, and both in-processing and post-processing approaches have been developed to satisfy these constraints (Dwork et al., 2012; Agarwal et al., 2018; Hardt et al., 2016; Zafar et al., 2019; Fitzsimons et al., 2019). Note that different fairness notions may be conflict with each other and which one to adopt is application and context dependent. In this work, we are interested in Equalized Loss (EL) fairness notion (Zhang et al., 2019; Berk et al., 2021) which implies that the prediction error should be the same across different groups,[2] and Group Bounded Loss (BGL) fairness notion (Agarwal et al., 2019) which requires the prediction error of every group to be bounded. We consider a relaxed version of EL fairness defined as follows.

**Definition 1 ($\gamma$-EL)** *A predictor $f$ satisfies $\gamma$-EL if the expected losses experienced by different demographic groups satisfy the following,*

$$-\gamma \leq L_0(\boldsymbol{w}) - L_1(\boldsymbol{w}) \leq \gamma. \tag{1}$$

Parameter $\gamma$ controls the degree of fairness; the smaller $\gamma$ implies the stronger fairness. When $\gamma = 0$, the exact EL fairness is attained. We say a group is *disadvantaged* if it experiences a larger loss. Similarly, Group Bounded Loss (BGL) fairness notion is formally defined as follows.

**Definition 2 ($\gamma$-BGL)** *A predictor $f$ satisfies $\gamma$-BGL if the expected loss of each demographic group is bounded by $\gamma$, i.e.,*

$$L_a(\boldsymbol{w}) \leq \gamma, \ \ \forall a \in \{0, 1\}. \tag{2}$$

---

[1]Predictive models such as logistic regression, linear regression, deep learning models, etc., are parameterized by a weight vector.

[2]EL has also been referred to as Overall Accuracy Equality in (Berk et al., 2021; Agarwal et al., 2019).

## 2.2 Relations between $\gamma$-EL and $\gamma$-BGL

In this section, we formally study the relations between $\gamma$-EL and $\gamma$-BGL fairness notions. Under $\gamma$-EL fairness constraint, finding a fair predictor is equivalent to solving the following constrained optimization problem:

$$\min_{\boldsymbol{w}} L(\boldsymbol{w}) \quad \text{s.t.} \quad |L_0(\boldsymbol{w}) - L_1(\boldsymbol{w})| \leq \gamma. \tag{3}$$

Let $\boldsymbol{w}^*$ be denoted as the solution to (3) and $f_{\boldsymbol{w}^*}$ is the optimal $\gamma$-EL fair predictor. Theorem 1 below shows that given the existence of a feasible point satisfying $\gamma$-BGL fairness, it's impossible for both groups experiencing loss larger than $\gamma$ from the optimal $\gamma$-EL fair predictor.

**Theorem 1** *Consider the following optimization for finding the optimal $\gamma$-BGL fair predictor,*

$$\min_{\boldsymbol{w}} L(\boldsymbol{w}) \ \text{s.t.} \ L_a(\boldsymbol{w}) \leq \gamma, \ \forall a \in \{0, 1\}. \tag{4}$$

*If $L_0(\boldsymbol{w}^*) > \gamma$ and $L_1(\boldsymbol{w}^*) > \gamma$, then optimization problem* (4) *does not have a feasible point.*

**Proof 1** *We prove by contradiction. Assume $\tilde{\boldsymbol{w}}$ is a feasible point of optimization* (4). *Note that $\tilde{\boldsymbol{w}}$ is a feasible point for optimization problem* (3) *as well. Since both $L_0(\boldsymbol{w}^*)$ and $L_1(\boldsymbol{w}^*)$ are larger than $\gamma$, we have,*

$$\begin{aligned}
\mathbb{E}\{l(Y, f_{\boldsymbol{w}^*})\} &= \Pr\{A = 0\}L_0(\boldsymbol{w}^*) + \Pr\{A = 1\}L_1(\boldsymbol{w}^*) > \gamma, \\
\mathbb{E}\{l(Y, f_{\tilde{\boldsymbol{w}}})\} &= \Pr\{A = 0\}L_0(\tilde{\boldsymbol{w}}) + \Pr\{A = 1\}L_1(\tilde{\boldsymbol{w}}) \leq \gamma.
\end{aligned}$$

*Therefore, $\boldsymbol{w}^*$ can not be the solution to* (3). *This contradiction proves that the optimization problem* (4) *cannot have a feasible point.*

Theorem 1 implies that if $\gamma$-EL notion leads to an increase of the loss of every demographic group, then there is no optimal predictor under $\gamma$-BGL.[3] The next theorem further shows that for any predictor satisfying $\gamma$-EL, it must satisfy $2\gamma$-BGL.

**Theorem 2** *Assume optimization problem* (4) *has at least one feasible point. Then, we have,*

$$\min\{L_0(\boldsymbol{w}^*), L_1(\boldsymbol{w}^*)\} \leq \gamma \ \text{and} \ \max\{L_0(\boldsymbol{w}^*), L_1(\boldsymbol{w}^*)\} \leq 2\gamma.$$

**Proof 2** *Let $\tilde{\boldsymbol{w}}$ be a feasible point of optimization problem* (4), *then $\tilde{\boldsymbol{w}}$ is also a feasible point to* (3). *If $\min\{L_0(\boldsymbol{w}^*), L_1(\boldsymbol{w}^*)\} > \gamma$, then $L(\boldsymbol{w}^*) > \gamma \geq L(\tilde{\boldsymbol{w}})$ must hold. This is a contradiction because it implies that $\boldsymbol{w}^*$ is not an optimal solution to* (3). *Therefore, $\min\{L_0(\boldsymbol{w}^*), L_1(\boldsymbol{w}^*)\} \leq \gamma$. Similarly, we can prove $\max\{L_0(\boldsymbol{w}^*), L_1(\boldsymbol{w}^*)\} \leq 2\gamma$ by contradiction. Assume $\max\{L_0(\boldsymbol{w}^*), L_1(\boldsymbol{w}^*)\} > 2\gamma$. Then, $\max\{L_0(\boldsymbol{w}^*), L_1(\boldsymbol{w}^*)\} - \min\{L_0(\boldsymbol{w}^*), L_1(\boldsymbol{w}^*)\} > \gamma$ which shows that $\boldsymbol{w}^*$ is not a feasible point for* (3). *This is a contradiction. Therefore, $\max\{L_0(\boldsymbol{w}^*), L_1(\boldsymbol{w}^*)\} \leq 2\gamma$.*

Theorems 1 and 2 investigated the relations between EL and BGL fairness notions. Since $\gamma$-EL implies $2\gamma$-BGL and it additionally requires the approximate equality across different groups, we will focus on $\gamma$-EL fairness notion in the rest of the paper. Because optimization problem (3) is a non-convex optimization, finding the optimal fair $\gamma$-EL solution efficiently can be challenging. In the next sections, we propose a number of algorithms that are easy to implement and can solve the optimization (3) efficiently.

## 3 Optimal Fair Model under EL fairness

In this section, we consider the optimization problem (3) under the EL fairness constraint. Note that this optimization problem is non-convex and finding the global optimal solution is difficult. However, we propose an algorithm which is able to find the solution to non-convex optimization (3) by solving a sequence of *convex* optimization problems. Before presenting the algorithm, we need to introduce two assumptions.

**Assumption 1** $L_0(\boldsymbol{w})$, $L_1(\boldsymbol{w})$, *and* $L(\boldsymbol{w})$ *are strictly convex functions in* $\boldsymbol{w}$.

---

[3]Theorem 1 is related to (Agarwal et al., 2019). In particular, they considered $\gamma$-BGL fairness and mentioned that the equalized loss fairness notion may increase the loss of both groups.

---

**Algorithm 1:** Function `ELminimizer`

---

1    `ELminimizer` $(\boldsymbol{w}_{G_0}, \boldsymbol{w}_{G_1}, \epsilon, \gamma)$:

2      $\lambda_{start}^0 = L_0(\boldsymbol{w}_{G_0})$

3      $\lambda_{end}^0 = L_0(\boldsymbol{w}_{G_1})$

4      Define $\tilde{L}_1(\boldsymbol{w}) = L_1(\boldsymbol{w}) + \gamma$

5      $i = 0$

6      **while** $\lambda_{end}^{(i)} - \lambda_{start}^{(i)} > \epsilon$ **do**

7        $\lambda_{mid}^{(i)} = (\lambda_{end}^{(i)} + \lambda_{start}^{(i)})/2$;

8        Solve the following convex optimization problem,

$$\boldsymbol{w}_i^* = \arg\min_{\boldsymbol{w}} \tilde{L}_1(\boldsymbol{w}) \ \text{ s.t. } \ L_0(\boldsymbol{w}) \le \lambda_{mid}^{(i)} \tag{5}$$

9        $\lambda^{(i)} = \tilde{L}_1(\boldsymbol{w}_i^*)$;

10       **if** $\lambda^{(i)} \ge \lambda_{mid}^{(i)}$ **then**

11         $\lambda_{start}^{(i+1)} = \lambda_{mid}^{(i)}$; $\lambda_{end}^{(i+1)} = \lambda_{end}^{(i)}$;

12       **end**

13       **else**

14         $\lambda_{end}^{(i+1)} = \lambda_{mid}^{(i)}$; $\lambda_{start}^{(i+1)} = \lambda_{start}^{(i)}$;

15       **end**

16       $i = i + 1$;

17      **end**

18      **Return** $\boldsymbol{w}_i^*$

---

**Example 1** *Consider a linear classifier $f_{\boldsymbol{w}}(\boldsymbol{X}) = \boldsymbol{w}^T \boldsymbol{X}$ with squared loss $l(Y, f_{\boldsymbol{w}}(\boldsymbol{X})) = (\boldsymbol{w}^T \boldsymbol{X} - Y)^2$. In this example, $\mathbb{E}\{l(Y, f_{\boldsymbol{w}}(\boldsymbol{X}))\} = \boldsymbol{w}^T \mathbb{E}\{XX^T\}\boldsymbol{w} - 2\mathbb{E}\{YX^T\}\boldsymbol{w} + \mathbb{E}\{Y^2\}$ is strictly convex in $\boldsymbol{w}$ if covariance matrix $\mathbb{E}\{XX^T\}$ is positive definite. Similarly, $L_a(\boldsymbol{w})$ is strictly convex if $\mathbb{E}\{XX^T | A = a\}$ is positive definite.*

Let $\boldsymbol{w}_{G_a}$ be the weight vector minimizing the loss associated with group $A = a$. That is,

$$\boldsymbol{w}_{G_a} = \arg\min_{\boldsymbol{w}} L_a(\boldsymbol{w}). \tag{6}$$

Since optimization problem (6) is an unconstrained convex optimization problem, $\boldsymbol{w}_{G_a}$ can be found efficiently by the first order condition or the gradient descent. We make the following assumption.

**Assumption 2** *We assume that the following holds,*

$$L_0(\boldsymbol{w}_{G_0}) \le L_1(\boldsymbol{w}_{G_0}) \text{ and } L_1(\boldsymbol{w}_{G_1}) \le L_0(\boldsymbol{w}_{G_1}).$$

---

**Algorithm 2:** Solving Optimization Problem (3)

---

**Input:** $\boldsymbol{w}_{G_0}, \boldsymbol{w}_{G_1}, \epsilon, \gamma$

1   $\boldsymbol{w}_{\gamma} = $ `ELminimizer`$(\boldsymbol{w}_{G_0}, \boldsymbol{w}_{G_1}, \epsilon, \gamma)$;

2   $\boldsymbol{w}_{-\gamma} = $ `ELminimizer`$(\boldsymbol{w}_{G_0}, \boldsymbol{w}_{G_1}, \epsilon, -\gamma)$;

3   **if** $L(\boldsymbol{w}_{\gamma}) \le L(\boldsymbol{w}_{-\gamma})$ **then**

4     |   $\boldsymbol{w}^* = \boldsymbol{w}_{\gamma}$;

5   **end**

6   **else**

7     |   $\boldsymbol{w}^* = \boldsymbol{w}_{-\gamma}$;

8   **end**

**Output:** $\boldsymbol{w}^*$

---

Assumption 2 implies that when a group experiences its lowest possible loss, it should not be the disadvantaged group. Under Assumption 2, given $\boldsymbol{w}_{G_0}$ and $\boldsymbol{w}_{G_1}$, Algorithm 1 with $\gamma = 0$ (i.e., function `ELminimizer`$(\boldsymbol{w}_{G_0}, \boldsymbol{w}_{G_1}, \epsilon, 0)$) finds the optimal 0-EL fair solution, where parameter $\epsilon > 0$ specifies the stopping criterion; as $\epsilon \to 0$, the output approaches to the optimal solution. Intuitively, Algorithm 1 solves non-convex optimization (3) by solving a sequence of convex and constrained optimization problems. If $\gamma > 0$, Algorithm 2 finds the optimal predictor under $\gamma$-EL using function `ELminimizer`.

The convergence of Algorithm 1 for finding the optimal 0-EL fair solution, and convergence of Algorithm 2 for finding the optimal $\gamma$-EL fair solution are proved in the following theorems.

**Theorem 3** *Consider sequences $\{\lambda_{mid}^{(i)}|i = 1, 2, \ldots\}$ and $\{\boldsymbol{w}_i^*|i = 1, 2, \ldots\}$ generated by Algorithm 1 when $\gamma = 0$, i.e., ELminimizer($\boldsymbol{w}_{G_0}, \boldsymbol{w}_{G_1}, \epsilon \to 0, 0$). Under Assumptions 1 and 2, we have,*

$$\lim_{i \to \infty} \boldsymbol{w}_i^* = \boldsymbol{w}^* \ and \ \lim_{i \to \infty} \lambda_{mid}^{(i)} = \mathbb{E}\{L(Y, f_{\boldsymbol{w}^*}(X))\}$$

*where $f_{\boldsymbol{w}^*}$ is the optimal 0-EL fair predictor.*

Similarly, we can prove the convergence for the approximate EL fairness when $\gamma \neq 0$.

**Theorem 4** *Assume that $L_0(\boldsymbol{w}_{G_0}) - L_1(\boldsymbol{w}_{G_0}) < -\gamma$ and $L_0(\boldsymbol{w}_{G_1}) - L_1(\boldsymbol{w}_{G_1}) > \gamma$. Then, as $\epsilon \to 0$, the output of Algorithm 2 goes to the optimal $\gamma$-EL fair solution $\boldsymbol{w}^*$.*

**Complexity Analysis:** The `While` loop in Algorithm 1 is executed for $\mathcal{O}(\log(1/\epsilon))$ times. Therefore, Algorithm 1 needs to solve a constrained convex optimization problem for $\mathcal{O}(\log(1/\epsilon))$ times. Note that constrained convex optimization problems can be efficiently solved via sub-gradient methods (Nedić & Ozdaglar, 2009), brier methods (Wright, 2001), stochastic gradient descent with one projection (Mahdavi et al., 2012), etc. For instance, Nedić & Ozdaglar (2009) introduces a sub-gradient method that finds the saddle point of the Lagrangian function corresponding to (5) and it converges at the rate of $\mathcal{O}(1/k)$ ($k$ is the number of iterations). Therefore, if $\epsilon$ is the maximum error tolerance for (5), the total time complexity of Algorithm 2 is $\mathcal{O}(1/\epsilon \log(1/\epsilon))$.

## 4 SUB-OPTIMAL FAIR MODEL UNDER $\gamma$-EL

In Section 3, we have shown that non-convex optimization problem (3) can be reduced to a sequence of convex constrained optimizations (5), and based on this we proposed an algorithm (Algorithm 2) that finds the optimal $\gamma$-EL fair predictor. However, the proposed algorithm still requires solving a convex constrained optimization in each iteration. In this section, we propose another algorithm which finds a *sub-optimal* solution to optimization (3) without solving constrained optimization in each iteration.

The algorithm consists of two phases in sequence: (1) finding two weight vectors by solving two *unconstrained* convex optimization problems; (2) generating a new weight vector satisfying $\gamma$-EL fairness with the two weight vectors found in the first phase. Because of the convexity, two unconstrained convex optimization problems in the first phase can be solved efficiently.

**Phase 1: Unconstrained optimization.** In this phase, we remove EL fairness constraint and first solve the following uncontrained optimization problem,

$$\boldsymbol{w}_O = \arg \min_{\boldsymbol{w}} L(\boldsymbol{w}) \tag{7}$$

Because $L(\boldsymbol{w})$ is strictly convex in $\boldsymbol{w}$, the above optimization problem can be solved efficiently using the gradient descent method. Predictor $f_{\boldsymbol{w}_O}$ is the optimal predictor without fairness constraint, and $L(\boldsymbol{w}_O)$ is the smallest overall expected loss that is attainable. Let $\hat{a} = \arg \max_{a \in \{0,1\}} L_a(\boldsymbol{w}_O)$, i.e., group $\hat{a}$ is the group that is disadvantaged under predictor $f_{\boldsymbol{w}_O}$. Then, for the disadvantaged group $\hat{a}$, we find $\boldsymbol{w}_{G_{\hat{a}}}$ by solving unconstrained optimization problem (6).

**Phase 2: Binary search to find the fair predictor.** For $\beta \in [0, 1]$, we define the followings,

$$
\begin{aligned}
g(\beta) &= L_{\hat{a}}\big((1 - \beta)\boldsymbol{w}_O + \beta\boldsymbol{w}_{G_{\hat{a}}}\big) - L_{1-\hat{a}}\big((1 - \beta)\boldsymbol{w}_O + \beta\boldsymbol{w}_{G_{\hat{a}}}\big); \\
h(\beta) &= L\big((1 - \beta)\boldsymbol{w}_O + \beta\boldsymbol{w}_{G_{\hat{a}}}\big),
\end{aligned}
$$

where function $g(\beta)$ can be interpreted as loss disparity between two demographic group under predictor $f_{(1-\beta)\boldsymbol{w}_O + \beta\boldsymbol{w}_{G_{\hat{a}}}}$, and $h(\beta)$ is the corresponding overall expected loss. Some properties of functions $g(.)$ and $h(.)$ are summarized in the following theorem.

**Theorem 5** *Under Assumptions 1 and 2, the followings hold,*

*1. There exists $\beta_0 \in [0, 1]$ such that $g(\beta_0) = 0$.*

*2. $h(\beta)$ is strictly increasing in $\beta \in [0, 1]$; $g(\beta)$ is strictly decreasing in $\beta \in [0, 1]$.*

Theorem 5 implies that in a $d_w$ dimensional space, if we start from $\boldsymbol{w}_O$ and move toward $\boldsymbol{w}_{G_{\hat{a}}}$ along a straight line, the overall loss increases and the disparity between two groups decreases until we reach $(1 - \beta_0)\boldsymbol{w}_O + \beta_0\boldsymbol{w}_{G_{\hat{a}}}$, at which 0-EL fairness is satisfied. Note that $\beta_0$ is the unique root of $g$. Since $g(\beta)$ is a strictly decreasing function, $\beta_0$ can be found using binary search. For the approximate $\gamma$-EL fairness, there are multiple values of $\beta$ such that $(1 - \beta)\boldsymbol{w}_O + \beta\boldsymbol{w}_{G_{\hat{a}}}$ satisfies $\gamma$-EL. Since $h(\beta)$ is strictly increasing in $\beta$, among all $\beta$ that satisfies $\gamma$-EL fairness, we would choose the smallest one. The method for finding a sub-optimal solution to optimization (3) is described in Algorithm 3.

---

**Algorithm 3:** Sub-optimal solution to optimization problem (3)

1   **Input:** $\boldsymbol{w}_{G_{\hat{a}}}, \boldsymbol{w}_O, \epsilon, \gamma$

2   Initialization: $g_\gamma(\beta) = g(\beta) - \gamma, i = 0, \beta_{start}^{(0)} = 0, \beta_{end}^{(0)} = 1$

3   **if** $g_\gamma(0) \leq 0$ **then**

4     |   $\underline{\boldsymbol{w}} = \boldsymbol{w}_O$, and go to line 16;

5   **end**

6   **while** $\beta_{end}^{(i)} - \beta_{start}^{(i)} > \epsilon$ **do**

7     |   $\beta_{mid}^{(i)} = (\beta_{start}^{(i)} + \beta_{end}^{(i)})/2$;

8     |   **if** $g_\gamma(\beta_{mid}^{(i)}) \geq 0$ **then**

9       |    |   $\beta_{start}^{(i+1)} = \beta_{mid}^{(i)}, \beta_{end}^{(i+1)} = \beta_{end}^{(i)}$;

10    |   **end**

11    |   **else**

12      |    |   $\beta_{start}^{(i+1)} = \beta_{start}^{(i)}, \beta_{end}^{(i+1)} = \beta_{mid}^{(i)}$;

13    |   **end**

14   **end**

15   $\underline{\boldsymbol{w}} = (1 - \beta_{mid}^{(i)})\boldsymbol{w}_O + \beta_{mid}^{(i)}\boldsymbol{w}_{G_{\hat{a}}}$;

16   **Output:** $\underline{\boldsymbol{w}}$

---

Note that `while` loop in Algorithm 3 is repeated for $\mathcal{O}(\log(1/\epsilon))$ times. Since the time complexity of operations in each loop is $\mathcal{O}(1)$, the total time complexity of Algorithm 3 is $\mathcal{O}(\log(1/\epsilon))$. We can formally prove that the output returned by Algorithm 3 satisfies $\gamma$-EL fairness constraint.

**Theorem 6** *Assume that Assumption 1 holds. If $g_\gamma(0) \leq 0$, then $\boldsymbol{w}_O$ satisfies the $\gamma$-EL fairness; if $g_\gamma(0) > 0$, then $\lim_{i \to \infty} \beta_{mid}^{(i)} = \beta_{mid}^{(\infty)}$ exists, and $(1 - \beta_{mid}^{(\infty)})\boldsymbol{w}_O + \beta_{mid}^{(\infty)}\boldsymbol{w}_{G_{\hat{a}}}$ satisfies the $\gamma$-EL fairness constraint.*

It is worth mentioning, since $h(\beta)$ is incrasing, we are intrested in finding the smallest possible $\beta$ that $(1 - \beta)\boldsymbol{w}_O + \beta\boldsymbol{w}_{G_{\hat{a}}}$ satisfies $\gamma$-EL. Here, $\beta_{mid}^{(\infty)}$ is the smallest possible $\beta$ under which $(1 - \beta)\boldsymbol{w}_O + \beta\boldsymbol{w}_{G_{\hat{a}}}$ satisfies $\gamma$-EL.

## 5   GENERALIZATION PERFORMANCE

So far we proposed algorithms for solving optimization (3). In practice, the joint probability distribution of $(\boldsymbol{X}, A, Y)$ is often unknown and the expected loss needs to be estimated using the empirical loss. Specifically, given $n$ samples $(\boldsymbol{X}_i, A_i, Y_i), i = 1, \ldots, n$ and predictor $f_{\boldsymbol{w}}$, the empirical losses of entire population and each group are defined as follows,

$$\hat{L}(\boldsymbol{w}) = \frac{1}{n} \sum_{i=1}^{n} l(Y_i, f_{\boldsymbol{w}}(\boldsymbol{X}_i)); \quad \hat{L}_a(\boldsymbol{w}) = \frac{1}{n_a} \sum_{i:A_i=a} l(Y_i, f_{\boldsymbol{w}}(\boldsymbol{X}_i)), \quad (8)$$

where $n_a = |\{i|A_i = a\}|$. Because $\gamma$-EL fairness constraint is defined in terms of expected loss, the optimization problem of finding an optimal $\gamma$-EL fair predictor using empirical losses is as follows,

$$\hat{\boldsymbol{w}} = \arg \min_{\boldsymbol{w}} \hat{L}(\boldsymbol{w}) \text{ s.t. } |\hat{L}_0(\boldsymbol{w}) - \hat{L}_1(\boldsymbol{w})| \leq \hat{\gamma}. \quad (9)$$

Note that $\hat{\gamma} \neq \gamma$ and one goal in this section is to find relation between $\hat{\gamma}$ and $\gamma$. We aim to investigate how to determine $\hat{\gamma}$ so that with high probability the predictor found by solving problem (9) satisfies

$\gamma$-EL fairness, and meanwhile $\hat{\boldsymbol{w}}$ is a good estimate of $\boldsymbol{w}^*$. To present our result, we make the following assumption.

**Assumption 3** *With probability $1 - \delta$, we have the following,*
$$\sup_{f_{\boldsymbol{w}} \in \mathcal{F}} |L(\boldsymbol{w}) - \hat{L}(\boldsymbol{w})| \leq B(\delta, n, \mathcal{F}),$$
*where $B(\delta, n, \mathcal{F})$ is a bound that goes to zero as $n$ goes to infinity.*

Note that if the class $\mathcal{F}$ is learnable with respect to loss function $l$, then there exists such a bound $B(\delta, n, \mathcal{F})$ that goes to zero as $n$ goes to infinity (Shalev-Shwartz & Ben-David, 2014).[4]

**Theorem 7** *Let $\mathcal{F}$ be a set of learnable functions, and let $f_{\hat{\boldsymbol{w}}}$ and $f_{\boldsymbol{w}^*}$ be the solution to (9) and (3) respectively with $\hat{\gamma} = \gamma + \sum_{a \in \{0,1\}} B(\delta, n_a, \mathcal{F})$. Then, with probability at least $1 - 6\delta$ the followings hold,*
$$L(\hat{\boldsymbol{w}}) - L(\boldsymbol{w}^*) \leq 2B(\delta, n, \mathcal{F}) \ \ and \ \ |L_0(\hat{\boldsymbol{w}}) - L_1(\hat{\boldsymbol{w}})| \leq \gamma + 2B(\delta, n_0, \mathcal{F}) + 2B(\delta, n_1, \mathcal{F}).$$

Theorem 7 shows that as $n_0$, $n_1$ go to infinity, $\hat{\gamma} \to \gamma$, and both empirical loss and expected loss satisfy $\gamma$-EL. In addition, as $n$ goes to infinity, the expected loss at $\hat{\boldsymbol{w}}$ goes to the minimum possible expected loss. Therefore, solving (9) using empirical loss is equivalent to solving (3) if the number of data points from each group is sufficiently large.

## 6 EXPERIMENTS

### 6.1 EXPERIMENT 1: QUADRATIC FUNCTIONS

First, we solve optimization problem (3) given the following quadratic functions,
$$\begin{aligned} L_0(\boldsymbol{w}) &= (w_1 + 5)^2 + (w_2 + 2)^2 + (w_3 + 1)^2 + 4w_1 \cdot w_3, \\ L_1(\boldsymbol{w}) &= (w_1 - 9)^2 + (w_2 - 9)^2 + (w_3 - 9)^2 + w_1 \cdot w_2 + w_2 \cdot w_3 + w_1 \cdot w_3 + 1, \\ L(\boldsymbol{w}) &= L_0(\boldsymbol{w}) + L_1(\boldsymbol{w}). \end{aligned}$$
By the first order condition, we obtain $\boldsymbol{w}_{G_0}, \boldsymbol{w}_{G_1}, \boldsymbol{w}_O$ as follows,
$$\boldsymbol{w}_{G_0} = [1, -2, -3]^T, \ \boldsymbol{w}_{G_1} = [4.5, 4.5, 4.5]^T, \ \boldsymbol{w}_O = [24.53, 3.0, 26.53]^T$$
We use Algorithm 1 to find the optimal solution to (3) and run Algorithm 3 to find a sub-optimal solution. In particular, we adopt the penalty method (Ben-Tal & Zibulevsky, 1997) to solve constrained convex optimization (5), i.e., by solving the following unconstrained optimization,
$$\min_{\boldsymbol{w}} L_1(\boldsymbol{w}) + t \cdot \max\{0, (L_0(\boldsymbol{w}) - \lambda_{mid}^{(i)})\}^2, \tag{10}$$
where $t$ is the penalty parameter. We solve the optimization problem (10) using gradient descent with learning rate 0.001 and 10000 iterations. We set penalty parameter $t = 0.5$ and increase $t$ by 0.1 after every 250 iterations. Note that optimization (5) is convex and the penalty method for a constrained convex optimization converges to the optimal solution (Ben-Tal & Zibulevsky, 1997).

We compare the our algorithms with a baseline: the solution to optimization problem (3) found using the penalty method, i.e., by solving the following unconstrained optimization,
$$\min_{\boldsymbol{w}} L_0(\boldsymbol{w}) + L_1(\boldsymbol{w}) + t \cdot \left[ \max\{0, (L_0(\boldsymbol{w}) - L_1(\boldsymbol{w}) - \gamma)\}^2 + \max\{0, (L_1(\boldsymbol{w}) - L_0(\boldsymbol{w}) - \gamma)\}^2 \right]. \tag{11}$$
When solving the optimization problem (11), we use learning rate 0.001. We set penalty parameter $t = 0.5$ and increase it by 0.1 every 250 iterations. Figure 1a illustrates the overall loss $L(\boldsymbol{w})$ at the (sub-) optimal points obtained from Algorithms 2 and 3 and the baseline. $x$-axis represents fairness parameter $\gamma$. Since Algorithm 2 converges to the optimal solution, it achieves the smallest loss. Figure 1b illustrates the distance of the optimal point $\boldsymbol{w}^*$ from the sub-optimal solutions obtained by Algorithm 3 and the baseline penalty method. It shows that when $\gamma$ is sufficiently large (less strict fairness constraint), a sub-optimal solution generated by Algorithm 3 is closer to the optimal solution than the solution found using the baseline method.

---

[4]As an example, if $\mathcal{F}$ is a compact subset of linear predictors in Reproducing Kernel Hilbert Space (RKHS) and loss $l(y, f(x))$ is Lipschitz in $f(x)$ (second argument), then Assumption 3 can be satisfied (Bartlett & Mendelson, 2002). Vast majority of linear predictors such as support vector machine and logistic regression can be defined in RKHS.

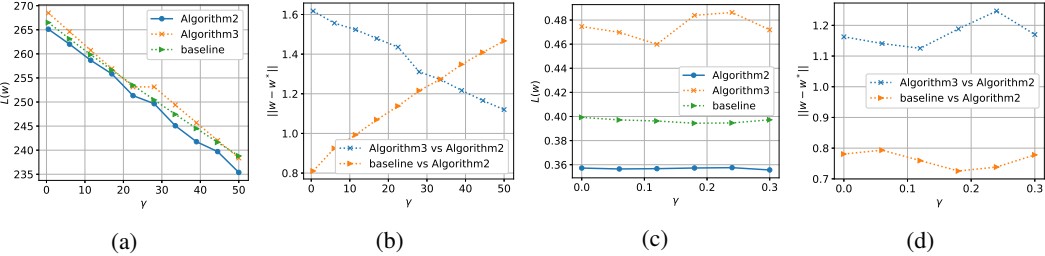

Figure 1: **a)** Experiment 1: loss as a function of fairness parameter $\gamma$. Algorithm 2 and Algorithm 3 significantly improve the loss compared to the baseline. **b)** Experiment 1: distance between the sub-optimal solution and the optimal solution. Algorithm 3 generates a sub-optimal solution closer to the optimal solution compared to the baseline. **c)** Experiment 2: loss as a function of fairness parameter $\gamma$. Both Algorithm 2 and Algorithm 3 outperform the baseline. **d)** Experiment 2: distance between the sub-optimal solution and the optimal solution.

## 6.2 EXPERIMENT 2: LOGISTIC REGRESSION AND THE ADULT INCOME DATASET

The adult income dataset is a public dataset containing the information of 48,842 individuals (Kohavi, 1996). Each data point includes 14 features including age, education, race, etc. Consider race (White or Black) as the sensitive attribute, we denote White demographic group by $A = 0$ and Black group by $A = 1$.

We first pre-process the dataset by removing the data points with a missing value or with the race other than Black and White and obtain 41,961 data points. Among these data points, 4585 belong to Black demographic group. For each data point, we convert all the categorical features to one-hot vectors and result in $d_x = 110$ dimensional features. We then normalize the feature vectors such that they have zero mean value and unit variance. Our goal is to find a logistic regression model satisfying $\gamma$-EL to predict whether the income of an individual is above \$50$K$ or not.

We use Algorithm 2 and Algorithm 3 with $\epsilon = 0.01$ to find the optimal logistic regression model under EL. We use the penalty method described in equation (11) as the baseline. Similar to Experiment 1, we set learning rate as 0.001 for solving (10) and (11). Penalty parameter $t$ is set to be 0.5 and increases by 0.1 every 250 iterations. Figure 1c illustrates the loss of logistic regression model trained by Algorithm 2, Algorithm 3, and the baseline. It shows that Algorithm 2 outperforms the baseline; this is because that the baseline only finds a sub-optimal solution while Algorithm 2 finds the global optimal solution. As mentioned in Section 4, Algorithm 3 finds a sub-optimal solution that satisfies $\gamma$-EL, and its performance can vary from case to case. Even though Algorithm 3 has a good performance in Experiment 1, it does not outperform the baseline in Experiment 2. Figure 1d illustrates the distances from the optimal point $\boldsymbol{w}^*$ to the sub-optimal solutions obtained by Algorithm 3 and the baseline penalty method. It shows that the distance from $\boldsymbol{w}^*$ to the solution obtained under Algorithm 3 is slightly larger than that from $\boldsymbol{w}^*$ to the solution obtained under the baseline.

## 7 CONCLUSION

In this work, we studied the problem of fair supervised learning under the Equalized Loss (EL) fairness notion which requires the prediction error/loss to be the same across different demographic groups. By imposing EL constraint, the learning problem can be formulated as a non-convex optimization problem. We introduce a number of algorithms that find the global optimal solution to this non-convex optimization problem. In particular, we showed that the optimal solution to such a non-convex problem can be found by solving a sequence of convex constrained optimizations. We also introduced a simple algorithm for finding a sub-optimal solution to the non-convex problem without solving constrained convex optimization problems. In addition to the theoretical guarantees, we demonstrated the performance of the proposed algorithm through numerical experiments.

## 8 REPRODUCIBILITY STATEMENT

Regarding the theoretical results: This paper includes six Theorems. The proof of Theorem 1 and Theorem 2 have been provided in the main text. Due to the page limit, the proofs of the other theorems have been provided in the appendix.

Regarding the numerical examples: the first experiment does not use any dataset, and we study the performance of our proposed method on quadratic objective functions. The values for hyper-parameters (including learning and penalty parameter) have been explicitly mentioned in section 6. In the second numerical example, we used the adult income dataset which is a well-known public dataset in our community. We explained the data pre-processing procedure in Section 6.2 in details.

## 9 ETHICS STATEMENT

In this work, we proposed algorithms to find fair predictors under the EL fairness notion. We want to emphasize that selecting a right fairness notion depends on the application and the authors do not make any suggestions to policy/law makers about choosing or avoiding this fairness notion.

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

APPENDIX

PROOFS

In order to prove Theorem 3, we first introduce two lemmas.

**Lemma 1** *Under assumption 2, there exists $\overline{\boldsymbol{w}} \in \mathbb{R}^{d_w}$ such that $L_0(\overline{\boldsymbol{w}}) = L_1(\overline{\boldsymbol{w}}) = L(\overline{\boldsymbol{w}})$ and $\lambda_{start}^{(1)} \le L(\overline{\boldsymbol{w}}) \le \lambda_{end}^{(1)}$.*

**Proof.** Let $h_0(\beta) = L_0((1 - \beta)\boldsymbol{w}_{G_0} + \beta\boldsymbol{w}_{G_1})$ and $h_1(\beta) = L_1((1 - \beta)\boldsymbol{w}_{G_0} + \beta\boldsymbol{w}_{G_1})$, and $h(\beta) = h_0(\beta) - h_1(\beta), \beta \in [0, 1]$. Note that $\nabla_{\boldsymbol{w}} L_a(\boldsymbol{w}_{G_a}) = 0$ because $\boldsymbol{w}_{G_a}$ is the minimizer of $L_a(\boldsymbol{w})$. Moreover, $\nabla_{\boldsymbol{w}}^2 L_a(\boldsymbol{w})$ is positive semi-definit because $L_a(.)$ is a strictly convex function.

First, we show that $L_0((1 - \beta)\boldsymbol{w}_{G_0} + \beta\boldsymbol{w}_{G_1})$ is an increasing function in $\beta$, and $L_1((1 - \beta)\boldsymbol{w}_{G_0} + \beta\boldsymbol{w}_{G_1})$ is a decreasing function in $\beta$. Note that $h_0'(0) = (\boldsymbol{w}_{G_1} - \boldsymbol{w}_{G_0})^T \nabla_{\boldsymbol{w}} L_0(\boldsymbol{w}_{G_0}) = 0$, and $h_0''(0) = (\boldsymbol{w}_{G_1} - \boldsymbol{w}_{G_0})^T \nabla_{\boldsymbol{w}}^2 L_0(\boldsymbol{w}_{G_0})(\boldsymbol{w}_{G_1} - \boldsymbol{w}_{G_0}) \ge 0$. This implies that $h_0'(\beta) \ge 0, \forall \beta \in [0, 1]$. Similarly, we can show that $h_1'(\beta) \le 0, \forall \beta \in [0, 1]$.

Note that under Assumption (2), $h(0) < 0$ and $h(1) > 0$. Therefore, by the intermediate value theorem, the exists $\overline{\beta} \in (0, 1)$ such that $h(\overline{\beta}) = 0$. Define $\overline{\boldsymbol{w}} = (1 - \overline{\beta})\boldsymbol{w}_{G_0} + \overline{\beta}\boldsymbol{w}_{G_1}$. We have,

$$h(\overline{\beta}) = 0 \implies L_0(\overline{\boldsymbol{w}}) = L_1(\overline{\boldsymbol{w}}) = L(\overline{\boldsymbol{w}}) \tag{12}$$

$$\boldsymbol{w}_{G_0} \text{is minimizer of } L_0 \implies L(\overline{\boldsymbol{w}}) = L_0(\overline{\boldsymbol{w}}) \ge \lambda_{start}^{(1)} \tag{13}$$

$$h_0'(\beta) \ge 0, \forall \beta \in [0, 1] \implies h_0(1) \ge h_0(\overline{\beta}) \implies \lambda_{end}^{(1)} \ge L_0(\overline{\boldsymbol{w}}) = L(\overline{\boldsymbol{w}}) \tag{14}$$

**Lemma 2** $L_0(\boldsymbol{w}_i^*) = \lambda_{mid}^{(i)}$, *where $\boldsymbol{w}_i^*$ is the solution to (5).*

**Proof.** We proceed by contradiction. Assume that $L_0(\boldsymbol{w}_i^*) < \lambda_{mid}^{(i)}$. Since $\boldsymbol{w}_{G_1}$ is not in the feasible set of (5), $\nabla_{\boldsymbol{w}} L_1(\boldsymbol{w}_i^*) \ne 0$. This is a contradiction because $\boldsymbol{w}_i^*$ is an interior point of the feasible set of a convex optimization and cannot be optimal if $\nabla_{\boldsymbol{w}} L_1(\boldsymbol{w}_i^*)$ is equal to zero.

**Proof [Theorem 3]**

Let $I_i = [\lambda_{start}^{(i)}, \lambda_{end}^{(i)}]$ be a sequence of intervals. It is easy to see that $I_1 \supseteq I_2 \supseteq \cdots$ and $\lambda_{end}^{(i)} - \lambda_{start}^{(i)} \to 0$ as $i \to \infty$. Therefore, by the Nested Interval Theorem, $\cap_{i=1}^{\infty} I_i$ consists of exactly one real number $\lambda^*$, and both $\lambda_{start}^{(i)}$ and $\lambda_{end}^{(i)}$ converge to $\lambda^*$. Because $\lambda_{mid}^{(i)} = \frac{\lambda_{start}^{(i)} + \lambda_{start}^{(i)}}{2}$, $\lambda_{mid}^{(i)}$ also converges to $\lambda^*$.

Now, we show that $L(\boldsymbol{w}^*) \in I_i$ for all $i$. Note that $L(\boldsymbol{w}^*) = L_0(\boldsymbol{w}^*) \ge \lambda_{start}^{(1)}$ because $\boldsymbol{w}_{G_0}$ is the minimizer of $L_0$. Moreover, $\lambda_{end}^{(1)} \ge L(\boldsymbol{w}^*)$ otherwise $L(\overline{\boldsymbol{w}}) < L(\boldsymbol{w}^*)$ ($\overline{\boldsymbol{w}}$ is defined in Lemma 1) and $\boldsymbol{w}^*$ is not optimal solution under 0-EL. Therefore, $L(\boldsymbol{w}^*) \in I_1$.

Now we proceed by induction. Suppose $L(\boldsymbol{w}^*) \in I_i$. We show that $L(\boldsymbol{w}^*) \in I_{i+1}$ as well. We consider two cases.

- $L(\boldsymbol{w}^*) \le \lambda_{mid}^{(i)}$. In this case $\boldsymbol{w}^*$ is a feasible point for (5), and $\lambda^{(i)} \le L(\boldsymbol{w}^*) \le \lambda_{mid}^{(i)}$. Therefore, $L(\boldsymbol{w}^*) \in I_{i+1}$.

- $L(\boldsymbol{w}^*) < \lambda_{mid}^{(i)}$. In this case, we proceed by contradiction to show that $\lambda^{(i)} \ge \lambda_{mid}^{(i)}$. Assume that $\lambda^{(i)} < \lambda_{mid}^{(i)}$. Define $g(\beta) = g_0(\beta) - g_1(\beta)$, where $g_i(\beta) = L_i((1 - \beta)\boldsymbol{w}_{G_0} + \beta\boldsymbol{w}_i^*)$. Note that $\lambda^{(i)} = g_1(1)$ By Lemma 2, $g_0(1) = \lambda_{mid}^{(i)}$. Therefore, $g(1) = \lambda_{mid}^{(i)} - \lambda^{(i)} > 0$. Moreover, under Assumption 2, $g(0) < 0$. Therefore, by the intermediate value theorem, there exists $\overline{\beta} \in (0, 1)$ such that $g(\overline{\beta}) = 0$. Similar to the proof of Lemma 1, we can show that $g_0(\beta)$ in an increasing function for all $\beta \in [0, 1]$. As a result $g_0(\overline{\beta}) <$

$g_0(1) = \lambda_{mid}^{(i)}$. Define $\overline{\boldsymbol{w}} = (1 - \overline{\beta})\boldsymbol{w}_{G_0} + \overline{\beta}\boldsymbol{w}_i^*$. We have,

$$g_0(\overline{\beta}) = L_0(\overline{\boldsymbol{w}}) = L_1(\overline{\boldsymbol{w}}) = L(\overline{\boldsymbol{w}}) < \lambda_{mid}^{(i)} \tag{15}$$

$$L(\boldsymbol{w}^*) < \lambda_{mid}^{(i)} \tag{16}$$

The last two equations imply that $\boldsymbol{w}^*$ is not an optimal fair solution under 0-EL fairness constraint. This is a contradiction. Therefore, if $L(\boldsymbol{w}^*) > \lambda_{mid}^{(i)}$, then $\lambda^{(i)} \geq \lambda_{mid}^{(i)}$. As a result, $L(\boldsymbol{w}^*) \in I_{i+1}$

By two above cases and the nested interval theorem, we conclude that,

$$L(\boldsymbol{w}^*) \in \cap_{i=1}^{\infty} I_i, \quad \lim_{i \to \infty} \lambda_{mid}^{(i)} = L(\boldsymbol{w}^*)$$

For the second part of the theorem, consider the following,

$$\boldsymbol{w}_{\infty}^* = \arg\min_{\boldsymbol{w}} L_1(\boldsymbol{w}) \, s.t., L_0(\boldsymbol{w}) \leq \lambda_{mid}^{\infty} = L(\boldsymbol{w}^*)$$

$$\lim_{i \to \infty} \boldsymbol{w}_i^* = \boldsymbol{w}_{\infty}^*$$

In order to show that $\boldsymbol{w}_{\infty}^*$ is equal to $\boldsymbol{w}^*$, we proceed by contradiction. Suppose $\boldsymbol{w}_{\infty}^* \neq \boldsymbol{w}^*$. As a result, $L_1(\boldsymbol{w}_{\infty}^*) < L(\boldsymbol{w}^*)$. Define $\eta(\beta) = \eta_0(\beta) - \eta_1(\beta)$, where $\eta_i(\beta) = L_i((1 - \beta)\boldsymbol{w}_{G_0} + \beta\boldsymbol{w}_{\infty}^*)$. Note that $L_1(\boldsymbol{w}_{\infty}^*) = \eta_1(1)$. By Lemma 2, the condition in (5) is binding and $\eta_0(1) = L(\boldsymbol{w}^*)$. Therefore, $\eta(1) = L(\boldsymbol{w}^*) - L_1(\boldsymbol{w}_{\infty}^*) > 0$. Moreover, under Assumption 2, $\eta(0) < 0$. Therefore, by the intermediate value theorem, there exists $\overline{\beta} \in (0, 1)$ such that $\eta(\overline{\beta}) = 0$. Similar to the proof of Lemma 1, we can show that $\eta_0(\beta)$ is an increasing function for all $\beta \in [0, 1]$. As a result $\eta_0(\overline{\beta}) < \eta_0(1) = L(\boldsymbol{w}^*)$. Define $\overline{\boldsymbol{w}} = (1 - \overline{\beta})\boldsymbol{w}_{G_0} + \overline{\beta}\boldsymbol{w}_{\infty}^*$. We have,

$$\eta_0(\overline{\beta}) = L_0(\overline{\boldsymbol{w}}) = L_1(\overline{\boldsymbol{w}}) = L(\overline{\boldsymbol{w}}) < L(\boldsymbol{w}^*) \tag{17}$$

The last equation implies that $\boldsymbol{w}^*$ is not an optimal fair solution under 0-EL fairness constraint. This a contradiction. As a result, $\boldsymbol{w}_{\infty}^* = \hat{\boldsymbol{w}}$.

**Proof [Theorem 4 ]**

Let $\boldsymbol{w}^*$ be the optimal weight vector under $\gamma$-EL.

**Step 1.** we show that one of the following holds,

$$L_0(\boldsymbol{w}^*) - L_1(\boldsymbol{w}^*) = \gamma \tag{18}$$

$$L_0(\boldsymbol{w}^*) - L_1(\boldsymbol{w}^*) = -\gamma \tag{19}$$

Proof by contradiction. Assume $-\gamma < L_0(\boldsymbol{w}^*) - L_1(\boldsymbol{w}^*) < \gamma$. This implies that $\boldsymbol{w}^*$ is an interior point of the feasible set of optimization problem (3). Since $\boldsymbol{w}^* \neq \boldsymbol{w}_O^*$, then $\nabla L(\boldsymbol{w}^*) \neq 0$. As a result, object function of (3) can be improved at $\boldsymbol{w}^*$ by moving toward $-\nabla L(\boldsymbol{w}^*)$. This a contradiction. Therefore, $|L_0(\boldsymbol{w}^*) - L_1(\boldsymbol{w}^*)| = \gamma$.

**Step 2.** Function $\boldsymbol{w}_{\gamma} = \texttt{ELminimizer}(\boldsymbol{w}_{G_0}, \boldsymbol{w}_{G_0}, \epsilon, \gamma)$ is the solution to the following optimization problem,

$$\min_{\boldsymbol{w}} \Pr\{A = 0\} L_0(\boldsymbol{w}) + \Pr\{A = 1\} L_1(\boldsymbol{w}), \, s.t., L_0(\boldsymbol{w}^*) - L_1(\boldsymbol{w}^*) = \gamma \tag{20}$$

To show the above claim, notice that the solution to optimization problem (20) is the same as the following,

$$\min_{\boldsymbol{w}} \Pr\{A = 0\} L_0(\boldsymbol{w}) + \Pr\{A = 1\} \tilde{L}_1(\boldsymbol{w}), \, s.t., L_0(\boldsymbol{w}^*) - \tilde{L}_1(\boldsymbol{w}^*) = 0, \tag{21}$$

where $\tilde{L}_1(\boldsymbol{w}) = L_1(\boldsymbol{w}) + \gamma$. Since $L_0(\boldsymbol{w}_{G_0}) - \tilde{L}_1(\boldsymbol{w}_{G_0}) < 0$ and $L_0(\boldsymbol{w}_{G_1}) - \tilde{L}_1(\boldsymbol{w}_{G_1}) > 0$, by Theorem 3, we know that $\boldsymbol{w}_{\gamma} = \texttt{ELminimizer}(\boldsymbol{w}_{G_0}, \boldsymbol{w}_{G_0}, \epsilon, \gamma)$ find the solution to (21).

Lastly, because $|L_0(\boldsymbol{w}^*) - L_1(\boldsymbol{w}^*)| = \gamma$, we have,

$$\boldsymbol{w}^* = \begin{cases} \boldsymbol{w}_\gamma & \text{if } L(\boldsymbol{w}_\gamma) \leq L(\boldsymbol{w}_{-\gamma}) \\ \boldsymbol{w}_{-\gamma} & \text{o.w.} \end{cases} \tag{22}$$

Thus, Algorithm 2 finds the solution to (3).

**Proof [Theorem 5]**

1. Under Assumption 2, $g(1) < 0$. Moreover, $g(0) \geq 0$. Therefore, by the intermediate value theorem, there exists $\beta_0 \in [0, 1]$ such that $g(\beta_0) = 0$.

2. Since $\boldsymbol{w}_O$ is the minimizer of $L(\boldsymbol{w})$, $h'(0) = 0$. Moreover, since $L(\boldsymbol{w})$ is strictly convex, $h''(0) > 0$. As a result, $h'(\beta) > 0$ for $\beta > 0$.

3. Since $\boldsymbol{w}_{G_{\hat{a}}}$ is the minimizer of $L_{\hat{a}}(\boldsymbol{w})$, and $L_{\hat{a}}(\boldsymbol{w})$ is strictly convex, $L_{\hat{a}}((1-\beta)\boldsymbol{w}_O + \beta\boldsymbol{w}_{G_{\hat{a}}})$ is strictly decreasing function.

   Note that since $h(\beta) = \Pr\{A = \hat{a}\}L_{\hat{a}}((1 - \beta)\boldsymbol{w}_O + \beta\boldsymbol{w}_{G_{\hat{a}}}) + \Pr\{A = 1 - \hat{a}\}L_{1-\hat{a}}((1 - \beta)\boldsymbol{w}_O + \beta\boldsymbol{w}_{G_{\hat{a}}})$ is strictly increasing and $L_{\hat{a}}((1 - \beta)\boldsymbol{w}_O + \beta\boldsymbol{w}_{G_{\hat{a}}})$ is strictly decreasing, we conclude that $L_{1-\hat{a}}((1 - \beta)\boldsymbol{w}_O + \beta\boldsymbol{w}_{G_{\hat{a}}})$ is strictly increasing. As a result, $g$ should be strictly decreasing.

**Proof [Theorem 6]** First, we show that if $g_\gamma(0) \leq 0$, then $\boldsymbol{w}_O$ satisfies $\gamma$-EL.

$$g_\gamma(0) \leq 0 \implies g(\beta) - \gamma \leq 0 \implies L_{\hat{a}}(\boldsymbol{w}_O) - L_{1-\hat{a}}(\boldsymbol{w}_O) \leq \gamma$$

Moreover, $L_{\hat{a}}(\boldsymbol{w}_O) - L_{1-\hat{a}}(\boldsymbol{w}_O) \geq 0$ because $\hat{a} = \arg\max_a L_a(\boldsymbol{w}_O)$. Therefore, $\gamma$-EL is satisfied.

Secondly, assume that $g_\gamma(0) > 0$. Under Assumption 1, $g_\gamma(1) = L_{\hat{a}}(\boldsymbol{w}_{G_{\hat{a}}}) - L_{1-\hat{a}}(\boldsymbol{w}_{G_{\hat{a}}}) - \gamma < 0$. Therefore, by the intermediate value there exists $\beta_0$ such that $g_\gamma(\beta_0) = 0$. Moreover, $g_\gamma$ is a strictly decreasing function. Therefore, the binary search proposed in Algorithm 3 converges to root of $g_\gamma(\beta)$. As a result, $(1 - \beta_{mid}^{(\infty)})\boldsymbol{w}_O + \beta_{mid}^{(\infty)}\boldsymbol{w}_{G_{\hat{a}}}$ satisfies satisfies $\gamma$-EL. Moreover, $L_{\hat{a}}(\boldsymbol{w}_O) - L_{1-\hat{a}}(\boldsymbol{w}_O) \geq 0$ because $\hat{a} = \arg\max_a L_a(\boldsymbol{w}_O)$. Note that since $g(\beta)$ is decreasing, $\beta_{mid}^{(\infty)}$ is the smallest possible $\beta$ under which $(1 - \beta)\boldsymbol{w}_O + \beta\boldsymbol{w}_{G_{\hat{a}}}$ $\gamma$-EL. Since $h$ is increasing, the smallest possible $\beta$ gives us a better accuracy.

**Proof [Theorem 7]**

By the triangle inequality, the following holds,

$$\sup_{f_{\boldsymbol{w}} \in \mathcal{F}} ||L_0(\boldsymbol{w}) - L_1(\boldsymbol{w})| - |\hat{L}_0(\boldsymbol{w}) - \hat{L}_1(\boldsymbol{w})|| \leq \sup_{f_{\boldsymbol{w}} \in \mathcal{F}} |L_0(\boldsymbol{w}) - \hat{L}_0(\boldsymbol{w})| + \sup_{f_{\boldsymbol{w}} \in \mathcal{F}} |L_1(\boldsymbol{w}) - \hat{L}_1(\boldsymbol{w})|. \tag{23}$$

Therefore, with probability at least $1 - 2\delta$ we have,

$$\sup_{f_{\boldsymbol{w}} \in \mathcal{F}} ||L_0(\boldsymbol{w}) - L_1(\boldsymbol{w})| - |\hat{L}_0(\boldsymbol{w}) - \hat{L}_1(\boldsymbol{w})|| \leq B(\delta, n_0, \mathcal{F}) + B(\delta, n_1, \mathcal{F}) \tag{24}$$

As a result, with probability $1 - 2\delta$ holds,

$$\{\boldsymbol{w}|f_{\boldsymbol{w}} \in \mathcal{F}, |L_0(\boldsymbol{w}) - L_1(\boldsymbol{w})| \leq \gamma\} \subseteq \{\boldsymbol{w}|f_{\boldsymbol{w}} \in \mathcal{F}, |\hat{L}_0(\boldsymbol{w}) - \hat{L}_1(\boldsymbol{w})| \leq \hat{\gamma}\} \tag{25}$$

Now consider the following,

$$L(\hat{\boldsymbol{w}}) - L(\boldsymbol{w}^*) = L(\hat{\boldsymbol{w}}) - \hat{L}(\hat{\boldsymbol{w}}) + \hat{L}(\hat{\boldsymbol{w}}) - \hat{L}(\boldsymbol{w}^*) + \hat{L}(\boldsymbol{w}^*) - L(\boldsymbol{w}^*) \tag{26}$$

By (25), $\hat{L}(\hat{\boldsymbol{w}}) - \hat{L}(\boldsymbol{w}^*) \leq 0$ with probability $1 - 2\delta$. Thus, with probability at least $1 - 2\delta$, we have,

$$L(\hat{\boldsymbol{w}}) - L(\boldsymbol{w}^*) \leq L(\hat{\boldsymbol{w}}) - \hat{L}(\hat{\boldsymbol{w}}) + \hat{L}(\boldsymbol{w}^*) - L(\boldsymbol{w}^*). \tag{27}$$

Therefore, under assumption 3, we can conclude with probability at least $1 - 6\delta$, $L(\hat{\boldsymbol{w}}) - L(\boldsymbol{w}^*) \leq 2B(\delta, n, \mathcal{F})$. In addition, by (24), with probability at least $1 - 2\delta$, we have,

$$
\begin{aligned}
|L_0(\hat{\boldsymbol{w}}) - L_1(\hat{\boldsymbol{w}})| \;&\leq\; B(\delta, n_0, \mathcal{F}) + B(\delta, n_1, \mathcal{F}) + |\hat{L}_0(\boldsymbol{w}) - \hat{L}_1(\boldsymbol{w})| \\
&\leq\; \hat{\gamma} + B(\delta, n_0, \mathcal{F}) + B(\delta, n_1, \mathcal{F}) = \gamma + 2B(\delta, n_0, \mathcal{F}) + 2B(\delta, n_1, \mathcal{F})
\end{aligned}
$$

