# OpenReview forum: "Non-convex Optimization for Learning a Fair Predictor under Equalized Loss Fairness Constraint"
_ICLR.cc/2022/Conference — ICLR 2022 Submitted_

### Official Review · Reviewer_Cpty · 2021-10-20

**Correctness:** 4
**Technical Novelty And Significance:** 3
**Empirical Novelty And Significance:** 4
**Recommendation:** 5
**Confidence:** 4

**Main Review:**

Strength:
- This paper considers a meaningful fairness notion EL, which is an important topic in fairness.
- This paper provides a complete analysis on how to handle the non-convex constrained fair learning problem. Their algorithms have provable guarantees.

Weakness:
- The theoretical proof is standard but seems not hard. There is no discussion on the technical contributions.
- The intuition of proposed algorithms seems to be dividing the constraint space into pieces, i.e., solving (5) for multiple \lambda_mid and then selecting an optimal one from satisfiable solutions. This idea seems also appeared in a prior paper [L. Elisa Celis, Lingxiao Huang, Vijay Keswani, Nisheeth K. Vishnoi: Classification with Fairness Constraints: A Meta-Algorithm with Provable Guarantees. FAT 2019: 319-328]. Could the authors provide a technical comparison?
- I have read the comment of Reviewer toHL. I agree that the missing conference (arXiv:1802.08626) is very important. The authors should provide a detailed comparison with this work.


**Summary Of The Paper:**

    This paper studies the problem of fair supervised learning under the Equalized Loss (EL) fairness notion, which is formulated as a non-convex constrained optimization problem. The authors introduce two algorithms that find the global (sub-)optimal solution by solving a sequence of convex (constrained) optimizations. Empirically, the algorithms perform well.

**Summary Of The Review:**

The paper considers an important fairness notion EL and provides algorithms with provable guarantees. Though their technique seems not hard, this may not be the main issue in the direction of fairness. Empirically, the proposed algorithms outperform baselines. The main weakness is the missing of several important prior works.

---

### Official Review · Reviewer_toHL · 2021-11-02

**Correctness:** 3
**Technical Novelty And Significance:** 3
**Empirical Novelty And Significance:** 1
**Recommendation:** 3
**Confidence:** 5

**Main Review:**

There is a well-known and closely related work in the literature, which is (arXiv:1802.08626). I am surprised that this paper was not cited in the paper. (arXiv:1802.08626) studies empirical risk minimization under fairness constraints as in this paper. (arXiv:1802.08626) also, propose to use equalized group loss as in this paper. See Definition 1 in (arXiv:1802.08626). (Their equalized loss definition is also conditioned on $y=1$ as in equal opportunity but I think their method can be also applied to equalized group loss that is defined as in this paper.) The similarities and the differences between the two papers should have been put clearly.
How does Theorem 7 be different than Theorem 1 in (arXiv:1802.08626)? Both theorems seem to be using the same assumption.
There are obvious similarities between the two papers.

I think the paper is very well-written, other than the fact that the related literature part should be improved. I am not convinced by the novelty of the paper given the existing work in (arXiv:1802.08626).
The contribution is limited in my point of view to the introduction of the algorithms to find the fair predictor, which is interesting but it is not enough to meet the barrier of acceptance.

The fact that the paper (arXiv:1802.08626) is not mentioned at all is worrying, given that even the notation of Assumption1 and Theorem 7 is the same with (arXiv:1802.08626).

The experiment section is far away from being complete. Only one real-world dataset is considered, while there are several real-world datasets publicly available and there is no comparison with the state-of-the-art.

In proof of Lemma 1, there is the gradient of L_a(w). But it could be the case that L is not smooth and thus L_a may not be smooth too. Then, instead of talking about the gradients, one should talk about the subgradients.
I am not sure in this case the proof of Lemma 2 goes through.
Minor comment: In the appendix, the {s.t.}'s should be put in math mode.

**Summary Of The Paper:**

This paper studies supervised learning models with fairness constraints.
They specifically consider equalized loss fairness constraint.
When a traditional (convex) loss minimization problem is cast with additional fairness constraints, the corresponding problem is non-convex.
They provide algorithms to efficiently solve this problem up to global optimality.
They demonstrate the performance of their algorithms on real-world data.


**Summary Of The Review:**

The contribution of the paper is not good enough given the existing literature on fair machine learning. It seems to me that this paper provides an extension to (arXiv:1802.08626). There are no numerical experiments to compare their framework with the state-of-the-art fair learning algorithms such as (arXiv:1802.08626, arXiv:1610.08452).

---

### Official Review · Reviewer_2B6v · 2021-11-02

**Correctness:** 4
**Technical Novelty And Significance:** 2
**Empirical Novelty And Significance:** 2
**Recommendation:** 3
**Confidence:** 3

**Details Of Ethics Concerns:**

The authors study how to optimize classification accuracy subject to a particular fairness constraint.

**Main Review:**

I found this paper to be interesting and well-motivated (finding the best classifier subject to fairness constraints is a real problem), but the results were not as compelling as I hoped. For instance, most of section 3 seems to stem from Assumption 1, wherein the loss functions L0, L1, and L are all assumed to be strictly convex. What happens if they are non-convex?

I would have appreciated some analysis of Algorithm 3’s performance — how much worse than Algorithm 2 can it be in theory? The results in the second experiment suggest that it can be quite a bit worse (even worse than the baseline).

There were also issues with writing quality, which is under the bar for ICLR. There were many missing articles (a/an/the) throughout the paper, and some of the exposition was very difficult to intuitively understand. Also, just a minor note, but expectation and probability are generally encased in [] or (), not {}, which is a convention I haven’t seen before.

Perhaps add a bit more motivation for the quadratic functions used in the first experiment (Section 6.1). How were they generated / chosen?

Also, just wondering: how likely is Assumption 2 to be satisfied in practice? Are there settings in which even the best loss for a certain group still means they're disadvantaged?


**Summary Of The Paper:**

The authors study fair prediction subject to Equalized Loss (EL), and they introduce a variety of approaches for exactly and approximately solving the problem of finding the globally optimal predictor that satisfies EL. First, they show how to solve a sequence of convex constrained optimization problems in order to solve the larger non-convex problem. Next, they show how to approximately solve this problem more efficiently by using unconstrained convex optimization. Lastly, they evaluate both of their approaches on two datasets.

**Summary Of The Review:**

I think the problem is a natural one (finding a good classifier subject to fairness constraints), but the technical results were not compelling and the writing quality was below the bar.

---

### Official Review · Reviewer_YrTG · 2021-11-06

**Correctness:** 4
**Technical Novelty And Significance:** 3
**Empirical Novelty And Significance:** 2
**Recommendation:** 6
**Confidence:** 3

**Main Review:**

Strengths:
- Theorem 1 and 2 are interesting. Bounded loss constrained minimization does not increase the maximum loss per group "too much" for classification problems. This can be a nice point to make to policy makers.
- The authors give a binary-search based thresholding method for optimizing bounded group loss constrained problems, which is also interesting, and might potentially have connections to existing optimization algorithms that use monotonicity of bounds to solve iterative subproblems.

Weaknesses:
- Assumptions: If the condition of feasibility of (4) in Theorem 2 is not true, then bounded increase in loss may not hold any more. For example, it is not true for facility location between two groups, where one can ask to minimize distance of a facility to each of two groups (for e.g., see, Too Many Fairness Metrics: Is There a Solution? By Gupta et.al. ). Placing a facility at infinity can make "absolute diff of group distances" to be zero, while there may be no facility that minimizes the total distances to both groups of populations. What happens in the case of classification losses? I suspect that the losses could then be unbounded. Which case might arise more frequently in practice? (if we want to think about relevance to policy).

- Assumption 2 implies that when a group experiences its lowest possible loss, it should not be the disadvantaged group. -- What is the rationale for this assumption? This might be misleading a bit -- as by disadvantaged the authors simply mean the non-optimized group, but since this is a "fairness" paper, it can also mean demographics groups which are at a disadvantage (minorities). Is Assumption 2 even needed, since the convex functions are assumed "strictly" convex?

- Related work - there is a lot of recent work on solving non-convex constrained formulations that result from "fair" objectives. To give a few examples:
1. Nonconvex optimization for regression with fairness constraints. http://proceedings.mlr.press/v80/komiyama18a/komiyama18a.pdf
2. Implicit Rate-Constrained Optimization of Non-decomposable Objectives http://proceedings.mlr.press/v139/kumar21b/kumar21b.pdf
3. Optimization with Non-Differentiable Constraints with Applications to Fairness, Recall, Churn, and Other Goals https://jmlr.csail.mit.edu/papers/volume20/18-616/18-616.pdf
4. Decomposition approach using monotonicity of parameters: e.g., Separable Convex Optimization with Nested Lower and Upper Constraints. https://pubsonline.informs.org/doi/10.1287/ijoo.2018.0004


How does the current work compare to these - in theory and experiments?

**Summary Of The Paper:**

The authors consider minimization of convex losses constrained by either bounded loss on each group, or bounded difference of losses over two groups. The second formulation is non-convex, whereas the first formulation is convex.

When the losses are strictly convex on both the demographic groups, so that their optima are distinct (I think this is the condition they need, but they use a more restrictive condition in the paper), they can find the "EL" fair predictor by solving a sequence of convex constrained optimizations, by exploiting a monotonicity property. They next give a more computationally efficient approximate algorithm for finding the EL fair predictor.

**Summary Of The Review:**

I like the way the authors approach the fairness constrained problem, and their study of property of feasible optimal solutions. However, I think the writing lacks justifications of the various assumptions, which makes the results very narrow. Further, there are many many results that consider the non-convex formulations resulting from minimum norm over demographic groups and composite objectives, and so though I think the algorithms proposed by the authors are nice, I suspect that these can be subsumed by existing literation in constrained optimization, and I would like the authors to comment on that.

---

### Decision · Program_Chairs · 2022-01-20

**Decision:**

Reject

**Comment:**

The paper considers learning classifiers under a fairness constraint which enforces the loss to be equal on certain subgroups. Reviewers found the work to be well-motivated, but raised concerns on the lack of discussion and comparison to relevant prior work. Notable examples in the fairness literature are Donini et al., "Empirical Risk Minimization under Fairness Constraints", Celis et al., "Classification with Fairness Constraints: A Meta-Algorithm with Provable Guarantees", while in the more broader constrained optimization literature, Kumar et al. "Implicit Rate-Constrained Optimization of Non-decomposable Objectives". The authors are encouraged to incorporate reviewers' detailed comments for a revised version of this work.